# A Novel Phantom for Standardized Microcalcification Detection Developed Using a Crystalline Growth System

**DOI:** 10.3390/tomography11030025

**Published:** 2025-02-27

**Authors:** Dee H. Wu, Caroline Preskitt, Natalie Stratemeier, Hunter Lau, Sreeja Ponnam, Supriya Koya

**Affiliations:** 1University of Oklahoma Health Sciences Center, Oklahoma City, OK 73104, USA; carlie.preskitt@gmail.com (C.P.); natalie-stratemeier@ouhsc.edu (N.S.); hunter-lau@ouhsc.edu (H.L.); supriya-koya@ouhsc.edu (S.K.); 2Kansas City School of Medicine, University of Missouri, Kansas City, MO 64108, USA; sreeyo.ponnam@gmail.com

**Keywords:** digital breast tomosynthesis (DBT), microcalcifications, phantom models, standardization, diagnostic accuracy

## Abstract

Background/Objectives: The accurate detection of microcalcifications in mammograms is critical for the early detection of breast cancer. However, the variability between different manufacturers is significant, particularly with digital breast tomosynthesis (DBT). Manufacturers have many design differences, including sweep angles, detector types, reconstruction techniques, filters, and focal spot construction. This study outlined the development of an innovative phantom model using crystallizations to improve the accuracy of imaging microcalcifications in DBT. The goal of these models was to achieve consistent evaluations, thereby reducing the variability between different scanners. Methods: We created a novel phantom model that simulates different types of breast tissue densities with calcifications. Furthermore, these crystalline-grown phantoms can more accurately represent the physiological shapes and compositions of microcalcifications than do other available phantoms for calcifications and can be evaluated on different systems. Microcalcification patterns were generated using the evaporation of sodium chloride, transplantation of calcium carbonate crystals, and/or injection of hydroxyapatite. These patterns were embedded in multiple layers within the wax to simulate various depths and distributions of calcifications with the ability to generate a large variety of patterns. Results: The tomosynthesis imaging revealed phantoms that utilized calcium carbonate crystals showed demonstrable visualization differences between the 3D DBT reconstructions and the magnification/2D view, illustrating the model’s value. The phantom was able to highlight changes in the contrast and resolution, which is crucial for accurate microcalcification evaluation. Conclusions: Based on the crystalline growth, this phantom model offers an important new standardized target for evaluating DBT systems. By promoting standardization, especially through the development of advanced breast calcification phantoms, this work and design aimed to contribute to improving earlier and more accurate breast cancer detection.

## 1. Introduction

A critical design objective of mammography scanners for breast radiologists is that the technology should provide them confidence in the accurate diagnostic evaluation of calcifications, a goal that can be achieved by the better validation of calcification detection during manufacturing. The correct identification of calcifications is essential for the early detection of breast cancer and ensures that patients receive accurate and reliable diagnostic evaluations. In addition, 3D digital breast tomosynthesis (DBT) is increasingly used for breast imaging; however, there is a notable trade-off in the visibility of microcalcifications when DBT is performed alone [1]. For this reason, DBT is used in combination with conventional 2D full-field digital mammography (FFDM), which can result in greater radiation doses to the patient. In this work, we used a novel set of phantom designs that add to the arsenal of potential tools that can provide accurate and reliable assessments of calcifications. Our design replicates clinical scenarios that involve microcalcifications, can potentially assist in developing better microcalcification evaluation, and is compatible with multiple vendors despite the technical differences in mammography equipment.

While mammography has a high sensitivity for detecting calcifications, its specificity is limited, leading to a significant number of false-positive results [2]. This often subjects women to unnecessary follow-up tests, causing additional emotional and physical burdens. Despite these challenges, mammography remains a critical tool in early breast cancer detection, with around 56,500 new DCIS cases and over 310,000 invasive breast cancer cases expected in 2024 [3].

Our study aimed to develop a standardized phantom model that mimics the unique properties of human breast tissue, which contains various abnormalities, including microcalcifications. Unlike conventional phantoms, our design leverages crystalline growth methods to create a more accurate representation of the complex shapes and compositions of microcalcifications. Our aim was to develop a breast phantom for image quality evaluations and comparison between scanners.

### 1.1. Design Considerations and Trade-Offs in Breast Tomosynthesis Technology

Breast tomosynthesis technology construction depends on various intricate design factors, including the focal size, exposure levels, motion type (i.e., continuous or stepped), sweep angle, detector choice, and reconstruction methods, all representing trade-offs that manufacturers must manage [4]. The resolution and contrast are distinct aspects of imaging; contrast often correlates with the X-ray exposure and dose received by the breast, while the resolution is influenced by the patient movement, sensor design, and number and angle of the tomosynthesis machine’s sweep, among other factors [5].

### 1.2. Sweep Angle Considerations in DBT: How Sweep Angle Impacts Affect the Resolution and Dose, Highlighting Key Differences Between Vendors for Breast Calcifications

One of the key differences between vendors is the angle of sweep in DBT systems, as shown in Figure 1. Changes in the sweep angle affect the visualization of objects and increase the number of ‘shots’, all while attempting to remain dose neutral (regarding tradeoffs between 2D-FFPM and DBT) [6]. Wider angles (with more exposure and/or shots—e.g., 50 degrees) can produce lesser in-plane resolution over a focused area but provide greater resolution over a specific area when combined with narrower angles. Thus, with the dose control assumption for fewer ‘shots’ and/or less exposure, narrow-angle DBT (for example, 15 degrees) systems produce a higher in-plane spatial resolution, making them theoretically better at visualizing small objects, like microcalcifications. Additionally, narrow-angle DBT has shorter scan times, resulting in sharper images with fewer motion artifacts. Conversely, wide-angle DBT systems have a better out-of-plane spatial resolution, which helps differentiate findings from overlapping tissue, making them better for visualizing masses and architectural distortions [7]. Some phantom studies support the use of narrow-angle DBT for evaluating calcifications, with some suggesting it detects small, subtle calcifications better than wider-angle DBT [8]. However, the choice of sweep angle is just one factor that influences microcalcification visualization. Other key factors include the materials used in the imaging system (e.g., filtration and focal spot), detector and system electronics, reconstruction methodology, and ability to provide magnification and spot imaging. Modifications to these factors, beyond the sweep angle, also affect microcalcification detection. Without thorough phantom studies or clinical trials, the most effective systems for evaluating calcifications remain uncertain. These factors, including materials, detectors, and reconstruction methods, are explored further in the following sections.

### 1.3. Magnification/Spot Compression Techniques for Improved Calcification and for Overlapping Tissue Visualization of the Breast Tissue

When indeterminate calcifications are suspected on the 3D DBT or FFDM, further diagnostic imaging could be obtained with magnification views to provide a more comprehensive evaluation of these lesions. If the DBT/FFDM images reveal any concerns after the patient has left following their screening exam, the patient may be asked to return to the breast care center for additional images with magnification. This causes anxiety and stress for some patients as they await a more definitive result. Magnification methods serve as an adjunctive technique that can improve the visualization of calcification by providing an additional focused exposure, as displayed in Figure 2 (a second set of scans acquired in addition to the standard tomosynthesis sweep). This technique allows for focused imaging for targeted suspicious areas, reducing the likelihood of missing small abnormalities. Magnification or spot imaging often involves variations in breast positioning, a decreased focal size, potentially longer exposure times, and a higher kilovolt peak (kVp). This includes a special magnification stand to elevate the breast above the detector and increase the distance between the breast and the detector. Switching to a smaller focal spot (if available on the scanner) can produce a magnification or spot compression view. These views can be used to evaluate the breast for fine structure details, such as microcalcifications.

In the magnification technique, the breast is brought closer to the X-ray source and further from the detector, effectively “zooming in” on the area of interest. Alternatively, a specific breast area can be assessed in spot compression by reducing the tissue overlap. Physically, additional tools, such as compression paddles or cones, can help spread the breast tissue and improve the image clarity. These tools may be smaller and more focused on the area of interest, enhancing the visualization. Magnification and spot compression methods may often require additional adjustments to the exposure settings, such as longer exposure times and higher kVp settings. Thus, regardless of whether DBT is initially performed, these extra imaging methods can improve the ability to visualize microcalcifications.

### 1.4. Technologies for Phantoms and Evaluation

Phantoms are made from radiological tissue-equivalent materials, such as polymethyl methacrylate (PMMA) or epoxy resins. These materials are designed to mimic the attenuation properties of human breast tissue so that the phantom behaves similarly to real breast tissue under X-ray imaging [9]. Inside the phantom, various elements simulate different breast tissues and abnormalities.

#### 1.4.1. Standard Evaluation Phantoms on the Market (ACR)

ACR provides specific guidelines for using phantoms in mammography quality control. These phantoms are constructed with nylon whiskers to represent fine linear structures that are surrogates for the system’s ability to visualize small, thread-like structures. Additionally, clusters of specks for masses within the phantom mimic calcifications and masses in breast tissue, allowing for a first-order evaluation of the imaging system’s performance. While the phantoms used for American College of Radiology (ACR) accreditation were designed before most DBT implementations, as illustrated in Figure 3 [10,11], they are still used to accredit DBT systems to calibrate and improve these 3D models. Vendors often follow the ACR standards for mammography, which are vital for ensuring acceptance and quality assurance, and consider this adequate evaluation. However, these structures lack the details our phantom aimed to show regarding the microcalcifications that radiologists seek to adequately diagnose and evaluate, and thus, are included in our new phantoms.

#### 1.4.2. A Specialized Adjunctive Swirled Phantom on the Market by Sun Nuclear

Sun Nuclear (formerly CIRS) offers the Model 020 BR3D Breast Imaging Phantom, which was designed for tomosynthesis to assess lesion detectability in a heterogeneous, tissue-equivalent background. These phantoms capture multiple image slices to reduce dense breast tissue overlap and improve target detection challenges. Model 020 includes six breast-equivalent slabs, each with unique swirl patterns to create diverse backgrounds and facilitate better evaluation of the architectural distortion. Each slab features a representation of microcalcifications, fibrils, and masses for further pathological assessment, and consists of two tissue-equivalent materials mimicking 100% adipose and 100% gland tissues “swirled” together in an approximate 50/50 ratio by weight. The swirling is intended to create spatial details in multiple planes for evaluation by pathology. The goal is for each slab to have a unique swirl pattern, allowing the phantom to be arranged to create multiple backgrounds, albeit potentially different ones between phantoms. However, while a step forward in technology, the phantom does not include the crystallization-grown component we implemented for this purpose [12].

To calibrate and improve these 3D models (i.e., DBT), mammography phantoms are utilized with both 2D and 3D models. These phantoms are imaged with both modalities and are used to further the predictive power of the 3D models. Currently, there are some limitations to this process. The phantoms used are only in certain shapes, which are not realistic regarding microcalcifications seen in in vivo scenarios. Additionally, the methods through which 3D models are calibrated vary from vendor to vendor. The purpose of our current publication was to describe a novel and unique calcification design that closely mimics the goals of microcalcification structures, making a more ‘stable’ target for evaluation. Our phantoms differed from the ACR and CIRS phantoms, as we used an advanced method that applied crystal growth with the ability to generate a shape that was realistic and served its purpose for microcalcifications.

## 2. Materials and Methods

We developed unique breast calcification phantom models in various shapes and designs. While expanding to different institutions is a future goal, our current focus is on establishing standards to guide manufacturers in optimizing scanner hardware, software, and AI-based reconstruction. Each design is based on the BI-RADS classifications [13] and potentially those from the Virtual Imaging Clinical Trial for Regulatory Evaluation (VICTRE) [14].

The basic phantom models were created using either paraffin wax or microcrystalline wax as the base structure representative of breast tissue, as shown in Figure 4. Phantoms that utilized either wax underwent the same production process since the difference in wax was based on modeling normal breast tissue vs. dense breast tissue. The phantom design was accomplished by melting paraffin wax at a minimum temperature of 65 °C or microcrystalline wax at a minimum temperature of 80 °C until a clear and homogeneous consistency was reached. The liquid wax was then poured into a 6 cm (diameter) by 4 cm (depth) silicone circular mold in various 1 cm increments. After the wax was allowed to cool and harden for 24 hours within the mold, freeform capillary-like channels were engraved into the surface of the wax using a 24-gauge needle or similar device. These channels can be carved on the surface of each incremental layer of wax, depending on the desired number of layers of embedded calcifications.

To model the calcification patterns outlined by desired structures, crystals composed of calcium carbonate, sodium chloride, or hydroxyapatite were each used in various phantom renditions. These calcification patterns were accomplished via (1) evaporation, (2) transplantation, and (3) injection (Figure 5). For the evaporation, 5 g of sodium chloride (Morton brand non-iodized salt) was dissolved in 240 mL of water and pipetted into the aforementioned channels carved into the wax. The model was then left in direct sunlight for the saline solution to evaporate until only the crystalline sodium chloride remained in the channels. The evaporation time varied between phantom renditions and was dependent on the room temperature and humidity. For the transplantation, small black dolomite rocks were submerged entirely in approximately 100 mL of distilled white vinegar and left in an open container to evaporate. Once the vinegar was evaporated, various sizes and shapes of aragonite crystals (a naturally occurring crystalline form of calcium carbonate that forms along the surface of the rocks) were transplanted with forceps into the channels carved in the wax, as well as intentionally dispersed in clusters on the surface of the wax. These crystals could be easily manipulated in size and ranged from a sub-mm size to nearly 1 cm. For the injection, a commercial solution of hydroxyapatite (Pulpdent Activa Bioactive Cement, Watertown, MA, USA) was directly injected into channels carved in the wax and intentionally dispersed on the surface of the wax in clusters with diameters approximately between 0.5 and 5 mm.

Depending on the desired depth of the calcification pattern, phantom renditions (models) were made with calcifications embedded throughout various layers of the wax. Crystals were either left exposed on the surface of the 4 cm wax mold or embedded incrementally in 1 cm layers of wax. To accomplish the latter, melted wax was poured to a depth of 1, 2, or 3 cm into the mold and allowed to dry for 24 h. The methods of evaporation, transplantation, and injection were used to arrange calcification patterns on the wax surface and then covered with additional layers of melted wax. To avoid disturbing the crystal patterns during the embedding process, the wax was allowed to partially cool to a gelatinous consistency before being poured onto an area adjacent to the crystals. The mold was gently swirled to encourage an even distribution and the settling of the wax and left to cool for another 24 h. This process was repeated to create different renditions in which there was any combination of 1 to 4 layers of calcifications distributed throughout layers of wax with a total maximum depth of 4 cm (Figure 6).

To illustrate the reproducibility of the phantom under different conditions, additional experiments were conducted to enable the visualization of similarities and differences through measurement and evaluation. This included running the experiment on two different units in our hospital to demonstrate that the results can be replicated across different units and across time. Historically, visual evaluations in accordance with the ACR Quality Control (QC) Manual are baseline evaluations conducted through visual inspection by medical physicists or assistants, who count visualized structures. For example, in a standard ACR QC measurement, having three or more specks visible in the correct locations is considered passing according to the ACR Digital Mammography Phantom Scoring Key. For the evaluation across time, we took measurements ~2 months apart (26 November 2024 (taken in Room 2) and 24 January 2025). Additionally, we took measurements on two different scanners in different rooms to allow for a comparison between measurements taken on two machines on the same day, i.e., Room 1 (24 January 2025) and Room 3 (24 January 2025).

## 3. Results

Of the phantom models created, we found that the evaporation of sodium chloride and transplantation of calcium carbonate crystals within both waxes produced the best tomosynthesis images. While hydroxyapatite was visible on the tomosynthesis, the 3D reconstruction of these images and the edge quality were not optimal, likely due to the composition and lower refractive index of hydroxyapatite. Hydroxyapatite is also a more expensive medium and poses difficulty in creating microstructures similar to those of malignant calcifications but may be used in future iterations to model uniform calcifications, such as milk calcium calcifications. Provided are several examples of results from our development and scanning, as shown in Figure 7 and Figure 8.

The wax materials used were found to have X-ray attenuation values similar to that of human breast tissue. Paraffin and microcrystalline wax materials appropriately model varying breast densities, as microcrystalline wax has a higher density and is comparable with dense breast tissue, which has much higher X-ray attenuation, whereas paraffin wax is comparable with the density and lower attenuation of normal fatty breast tissue. In terms of handling, paraffin wax is cost-effective and easy to use but can be brittle and less durable, as shown in Figure 9. Microcrystalline wax, while more expensive, offers better flexibility, adhesion, and durability, making it potentially more suitable for breast imaging phantoms.

Quality visualization of the embedded crystals in the wax was performed and is shown in Figure 10, which reveals that the embedded crystals within the wax were clearly visualized in a comparison between a magnification view and a tomosynthesis reconstruction. These images were captured using a Selenia Dimensions scanner, magnification technique, and a single projected slice from the tomosynthesis.

Thus, the applications of these methods are either comparable to or exceed the baseline goals of the ACR QA phantom, particularly regarding the evaluation of calcifications. The physiological reasons for this are described in our Appendix A. We achieved a strong visual similarity of the results across time, taking measurements ~2 months apart (26 November 2024 and 24 January 2025 measurements). All three measurements in the magnification scans were comparable and with a decrease in quality, as shown on the DBT images. Additionally, by taking two histogram evaluations in the area of calcification on two different scanners, we were able to observe the similarities between the measurements from Room 1 and Room 3, as revealed by comparing Figure 11 panels B and D with Figure 11 panels C and E. We performed tests on the normalized histogram data (Figure 11D,E). To quantitate, we performed a two-sample Kolmogorov–Smirnov test and demonstrated that these measurements appeared to be drawn from the same distribution (test statistic for KS two-sample test: D* = 0.0328 to 0.1389 for the comparison, which indicates that the samples came from the same distribution), with visualizations of the histograms from recent measurements taken on 24 January 2025 conducted in MATLAB R2023A, Natick MathWorks.

Mathematically, the two-sample Kolmogorov–Smirnov (KS) test compares the cumulative distribution functions (CDFs) of two samples, as shown in Figure 12. The null hypothesis is that the samples likely originated from the same underlying distribution [15]. This measurement is more stringent than the goal of achieving similar visibility but shows what is possible and adds quantification beyond what the ACR performs. Additionally, the measurement from two months earlier shows a very similar appearance without a statistical difference. As expected, when we repeated the tomosynthesis images (Figure 11F), they showed consistent and poor visualization of calcifications. While we were not aiming for significant quantitative differences between these time points, we sought to illustrate that the images had high similarity in appearance, thus validating the work.

## 4. Discussion

This work reviews the challenges in developing and applying mammography scans for breast cancer detection, focusing on microcalcification visualization, which is crucial for early detection. Addressing these challenges is essential for improving imaging technology development in the area of DBT. Variations in DBT design highlight the need for standardized phantom targets to improve the microcalcification consistency. These variations, including differences in the detector types, reconstruction techniques, and filters (Table 1), underscore the importance of standardization. In this work, we report on a new phantom model that can move manufacturers toward standardized goals. Achieving a balance between image quality and radiation dose is crucial for advancing the standardized detection of critical pathologies, like breast calcifications. Current phantoms (ACR and CIRS) fall short in replicating real-world details, which we address with a novel phantom design that more accurately mimics microcalcifications in various shapes and sizes. Clinical trials in radiology, particularly on microcalcifications, are costly due to the number of diagnostic readers and the time required. Our phantoms offer a critical first step, closely resembling the pathology of interest, and can help standardize readings, making trials more cost-effective. Given the complexity of design objectives, achieving the right balance between image quality and radiation dose in manufacturing is essential to creating standardized tools for detecting key pathologies, like breast calcifications [16]. This advancement aimed to enhance the development of more accurate and reliable diagnostic tools and provide potentially significant aid, particularly in the age of artificial intelligence (AI)-based reconstruction. The phantom in this work demonstrated significantly improved contrast control and resolution, which can be critical for evaluating microcalcifications related to breast cancer.

Cognizant of innovations and concerns from the FDA, we recognize that in silico models can be used to study cases and test prototypes that would be difficult or impractical to study with actual patients. Badano et al. suggested utilizing in silico models to conduct simulated clinical trials of medical imaging systems, such as the VICTRE trial [17]. An extension of our work could utilize phantoms that can be designed to incorporate in silico models to provide insights into how patient characteristics influence the performance of different imaging technologies, thus combining the benefits these in silico methods provide with our calcification-based phantoms so as to realize them in actual machine systems [18]. Our technology aligns with the FDA’s perspective, in which computer-based modeling can improve device validation, facilitate smaller clinical trials, and allow us to gain valuable insights from simulations. In our case, we took this a step further; we could take their synthetic cases and render them into real-world models for calcifications using our methodology.

A critical aspect of this work is the use of phantoms with crystalline growth for mammography quality control, which directly benefits patients by improving the accuracy of breast cancer detection. By comparing the ACR standard phantom with specially designed phantoms for DBT, we highlighted the limitations of the ACR phantom and how advancements can lead to better imaging. These phantoms help ensure that breast cancer screenings are more reliable, in turn offering patients greater peace of mind through the improved detection of microcalcifications, which are crucial for early diagnosis. While ACR phantoms provide a basic framework for standardization, they were developed before DBT, and our improved phantoms more accurately reflect the shapes and compositions of real-life calcifications. This innovation can enhance the quality of care that healthcare institutions provide, ultimately leading to better outcomes and fewer unnecessary callbacks for patients.

As we enhance phantoms to better represent the constituent properties of actual biological tissues and features such as calcifications, we create new opportunities for standardization and validation efforts that benefit everyone involved. It is important to establish a comparison target that includes phantoms to help standardize image quality. We validated the phantom using different clinical imaging systems, ensuring artifact-free images comparable with those obtained from real breast tissues, which is a critical step for translational research emphasized by Saleh et al. [19]. Validation experiments that compared imaging systems in different clinical settings further confirmed the phantom’s reliability. To address the patient variability, we customized the phantom to represent diverse anatomies and tissue compositions, creating multiple versions with varying complexity and tissue interfaces, as outlined by Zhang and Fu [20]. This customization enhanced the phantom’s applicability across a broad spectrum of clinical scenarios. Much of the calibration and work has been guided by the seminal work of White et al. on epoxy resin [21]. We do not see this as a limitation of the experiment and recognize that the calibration of stoichiometric-based formulations is important for long-term calibration. The mimicry of glandular tissue has been a concern in our field and can be added to our models in the future. The exact formulation of BR-12 is proprietary, but it is generally made from an epoxy resin that is carefully engineered to mimic the X-ray attenuation properties of human breast tissue. Overlapping structures in the third dimension are beneficial for detecting architectural distortion, but given the size of calcification, this is less of an issue, as the in-plane resolution is needed to resolve the tiny structures. In this work, we present a more physiologically realistic model than crushed crystals. Our phantom technology was designed to closely replicate the physiological characteristics of breast microcalcifications, providing a more realistic basis for evaluating imaging performance. Additionally, our studies expanded the quantitative evaluation and consistency of calcifications across different scanners. We additionally demonstrated strong visual similarity in the results taken two months apart, showing consistent measurements. Furthermore, we were able to compare the results between different scanners and rooms, highlighting the robustness and reliability of our methods. Indeterminate cases during readings can lead to callbacks and significant patient stress, and for many women, this is a particularly troubling time, marked by uncertainty and fear. The anxiety associated with the waiting period frequently turns out to be more distressing than the actual results of the tests, regardless of whether they are positive or negative [22].

Variations in mammography system technology to visualize microcalcifications can lead to inconsistent interpretations and patient management. The complexity of visualizing and diagnosing breast calcifications highlights the need for standardized mammography technologies, enabling radiologists to achieve greater diagnostic accuracy. Providing physicians with consistently high-quality images across technologies, vendors, and institutions is crucial for accurate diagnoses, efficient workflows, and optimal patient care. This approach is especially important, as AI is increasingly integrated into diagnostic processes, where consistent image quality is essential for accurate analysis and patient care. This standardization may be particularly crucial not only for basic systems before the introduction of AI but also during its integration.

These guidelines typically encompass a range of competing objectives, including image quality, radiation dose, equipment performance, and quality control procedures. Our work focused on using phantoms designed with crystalline growth for quality control in mammography. Phantoms are essential tools for manufacturers and medical physicists seeking to enhance their design processes and increase confidence in imaging results. By comparing the ACR standard phantom with our specially designed digital breast tomosynthesis (DBT) phantoms, we highlighted the limitations of the ACR phantom. We investigated the technological aspects of DBT to understand how these components impacted image quality, particularly the visualization of microcalcifications, which we visually summarize in Figure 13.

## 5. Conclusions

Obtaining high-quality images of breast calcifications is a crucial responsibility for enhancing mammography in breast cancer detection. Currently, while the ACR phantoms provide a solid foundation for standardization using simple designs to replicate calcifications—such as materials like nylon whiskers—they were developed before the introduction of DBT. In contrast, our improved phantoms were designed with crystalline growth for quality control in mammography, allowing for a more accurate representation of the physiological shapes and compositions of actual calcifications, thereby enhancing standardization. Our advanced phantom model promotes uniformity in interpreting calcifications by mirroring real clinical conditions, enabling radiologists to evaluate calcifications consistently. By integrating these translational elements, our crystal phantom serves as a robust tool for standardizing mammography techniques and bridging the gap between imaging advancements and clinical practice. This approach not only boosts diagnostic accuracy but also aids in developing more patient-centered breast cancer screening protocols. These new phantom models can create opportunities for enhanced collaboration between vendors, device evaluators, and users, fostering a more unified approach.

## 6. Patents

Note: Patents are pending for this work, and the provisional patents are cited [18]. The issue fee for the patent application (OU 2021-025) was processed on 27 November 2024 and was finalized on 14 January 2025.

## Figures and Tables

**Figure 1 tomography-11-00025-f001:**
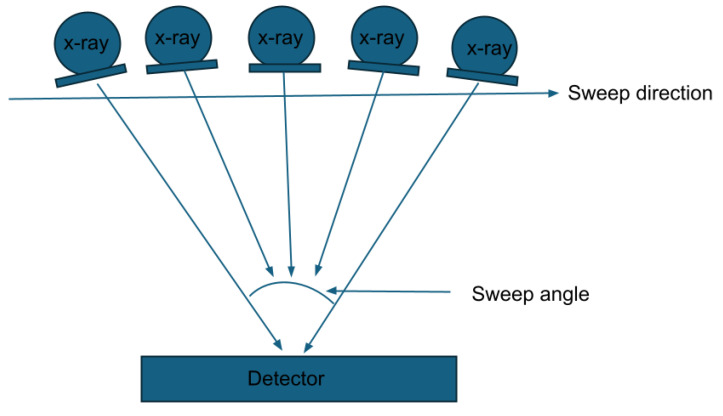
Sweep angle in digital breast tomosynthesis. Wider sweep angles can visualize larger areas with a lower resolution, while narrower angles provide more resolution over a smaller area, potentially improving the visualization of microcalcifications. This tradeoff and other factors, such as filtration, focal spot, detector, and reconstruction methods, require thorough phantom tests to determine the most effective DBT system for microcalcification evaluation.

**Figure 2 tomography-11-00025-f002:**
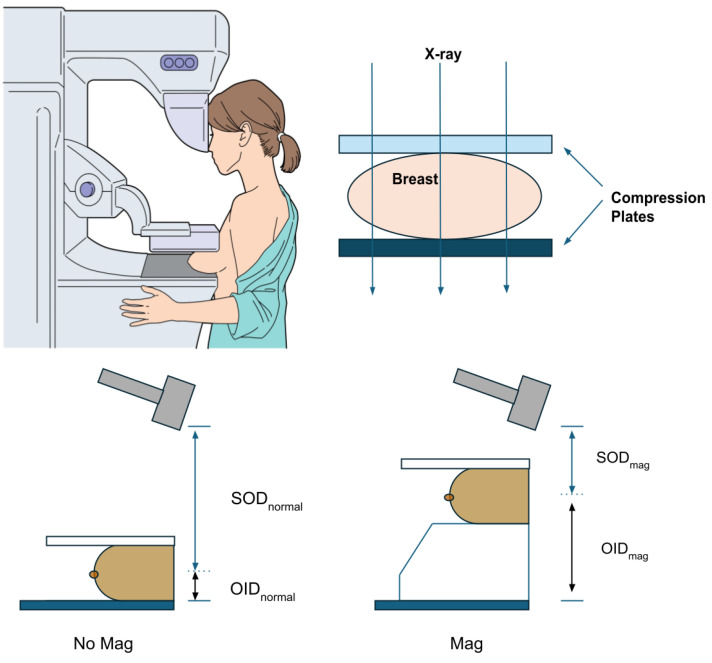
Magnification and spot compression in DBT. Magnification and spot compression techniques improve the visualization of microcalcifications in DBT. Magnification increases the distance between the breast and the detector. SOD—subject-to-object distance, OID—object-to-image distance.

**Figure 3 tomography-11-00025-f003:**
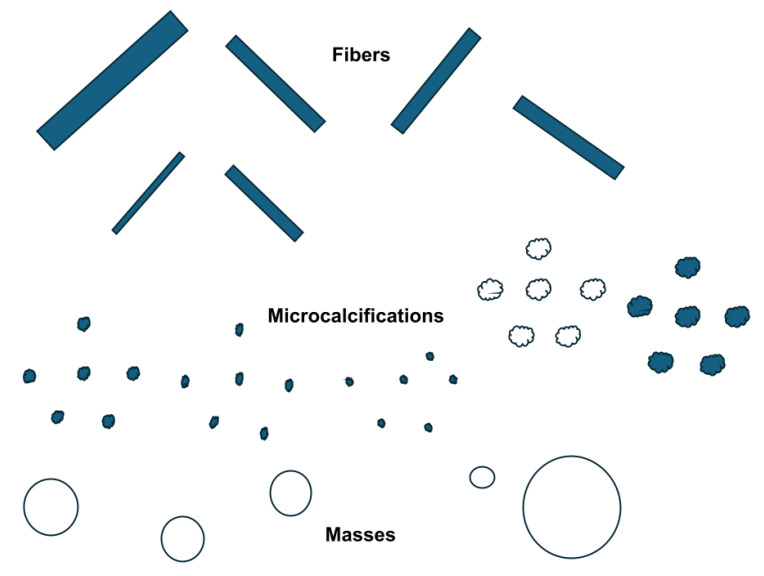
The American College of Radiology (ACR) phantom is used for quality control in mammography systems, including digital breast tomosynthesis (DBT). The phantom also contains rudimentary elements that mimic breast tissue and abnormalities, like microcalcifications and masses. The ACR guidelines ensure consistent and reliable image quality across different mammography systems.

**Figure 4 tomography-11-00025-f004:**
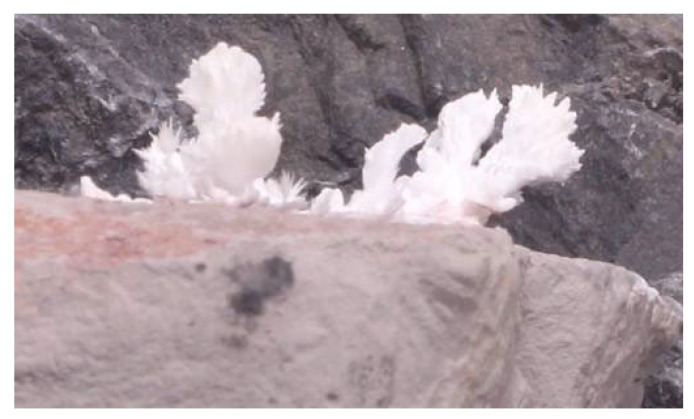
Crystals with ~0.1 mm resolution were successfully grown using standard dolomite rock and evaporation techniques, with potential for further enhancement through advanced laboratory methods.

**Figure 5 tomography-11-00025-f005:**
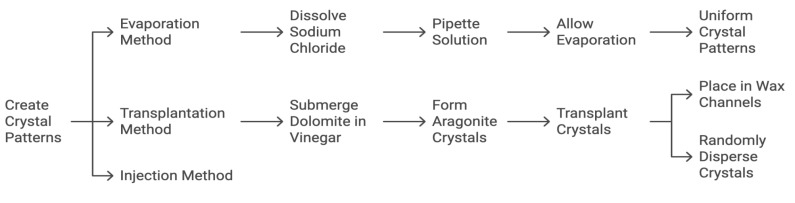
This flow diagram illustrates the steps for each of the three methods: evaporation, transplantation, and injection.

**Figure 6 tomography-11-00025-f006:**
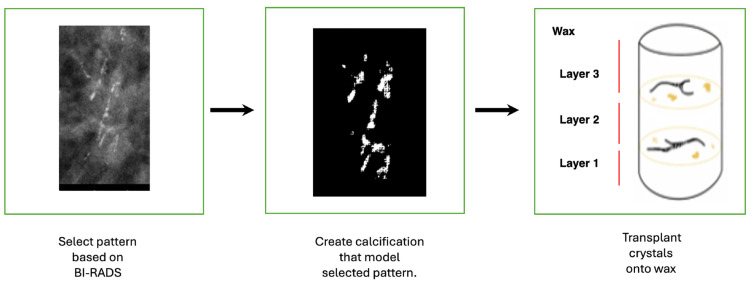
This flow diagram details the steps of modeling the phantom. The third image depicts the stacked calcification patterns between three layers of wax. This creates more dimensions within the phantom and can more accurately model widely dispersed calcifications as opposed to isolated calcifications.

**Figure 7 tomography-11-00025-f007:**
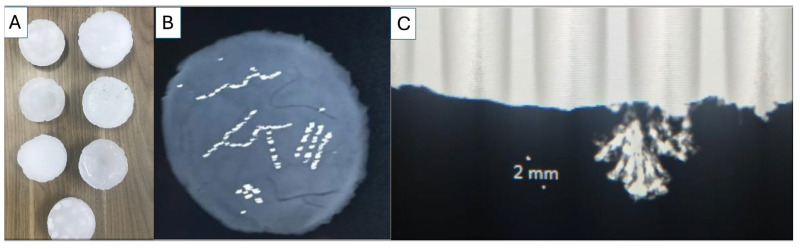
(**A**) Models of wax phantoms designed to simulate microcalcification detection in DBT imaging. Each phantom features variations in the density, pattern, size, and composition of calcifications. By providing a standardized model compatible across various mammography systems, these phantoms support consistent, accurate assessments of calcifications, especially in cases involving subtle or complex calcification shapes. (**B**) Tomosynthesis image of sodium chloride crystals arranged in various patterns, such as heterogeneous clusters and fine linear branches. (**C**) Radiographic image of a control calcium carbonate crystal grown on dolomite rock and measured to determine whether it appropriately modeled the size and shape of a malignant calcification modeled in BI-RADS.

**Figure 8 tomography-11-00025-f008:**
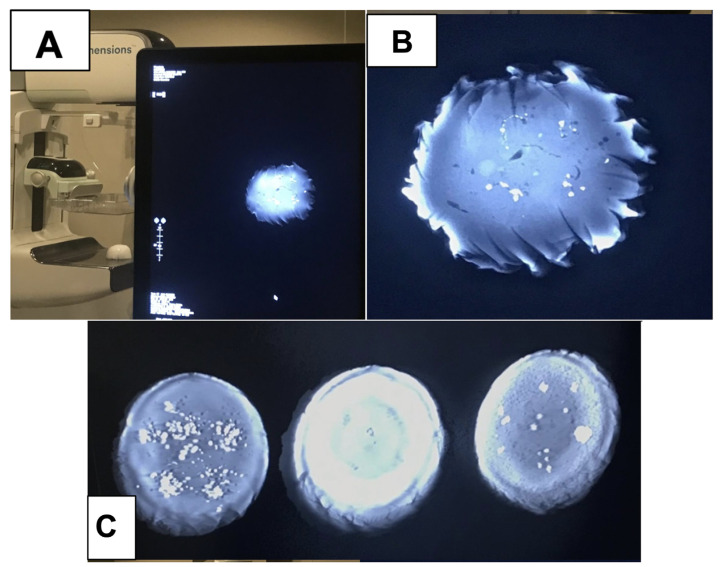
(**A**) A phantom being scanned using DBT. (**B**) A detailed image of the phantom being scanned. This paraffin wax phantom contains NaCl calcifications in various distributions, including amorphous and fine-branched linear patterns. (**C**) Three phantom renditions made with calcium chloride crystals dispersed in differing calcification patterns within microcrystalline wax.

**Figure 9 tomography-11-00025-f009:**
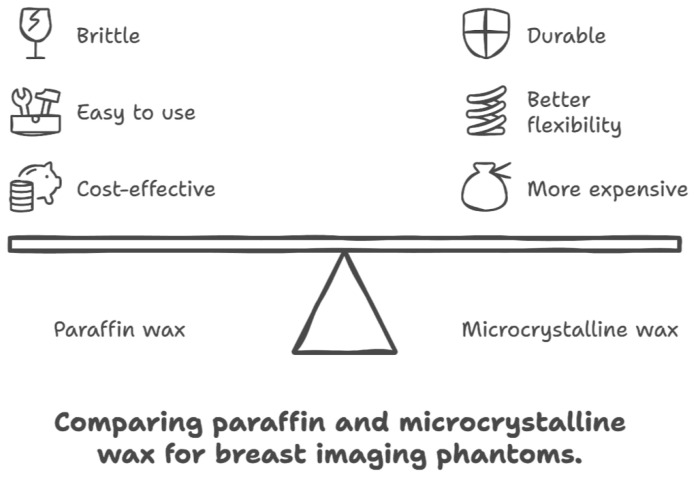
Comparison of costs and benefits between microcrystalline and paraffin wax when considering large-scale production of wax phantoms for future implications.

**Figure 10 tomography-11-00025-f010:**
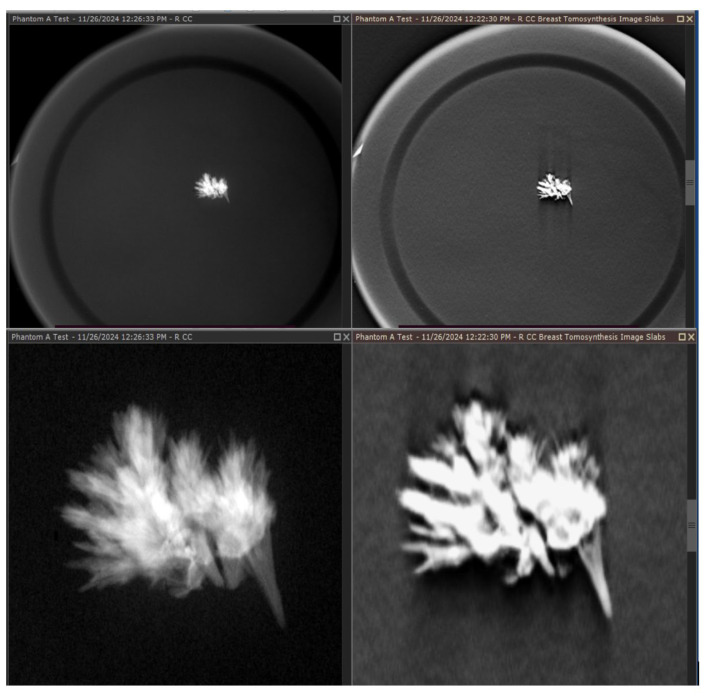
Magnification mode of our phantom image is shown on the left. The tomosynthesis image is shown on the right; the tomosynthesis image of the calcium carbonate embedded within the microcrystalline wax to model calcification in the breast was performed in a phantom and could be replicated on different machines during the development and evaluation of the systems.

**Figure 11 tomography-11-00025-f011:**
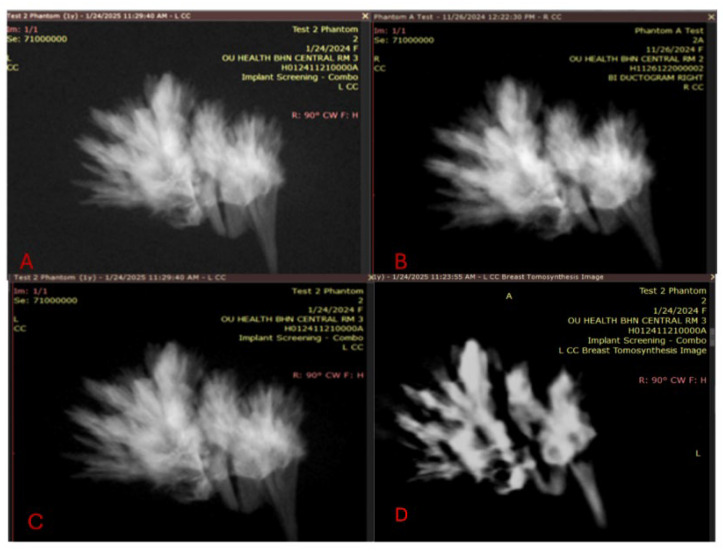
(**A**) Performed on 26 November 2024 in Room 2. (**B**) Repeat test performed on 24 January 2025 2 months later in Room 1. (**C**) In Room 3. (**D**) Tomosynthesis image taken from Room 3 on 24 November 2024. (**E**) Normalized histogram from Room 1. (**F**) Normalized histogram from Room 3.

**Figure 12 tomography-11-00025-f012:**
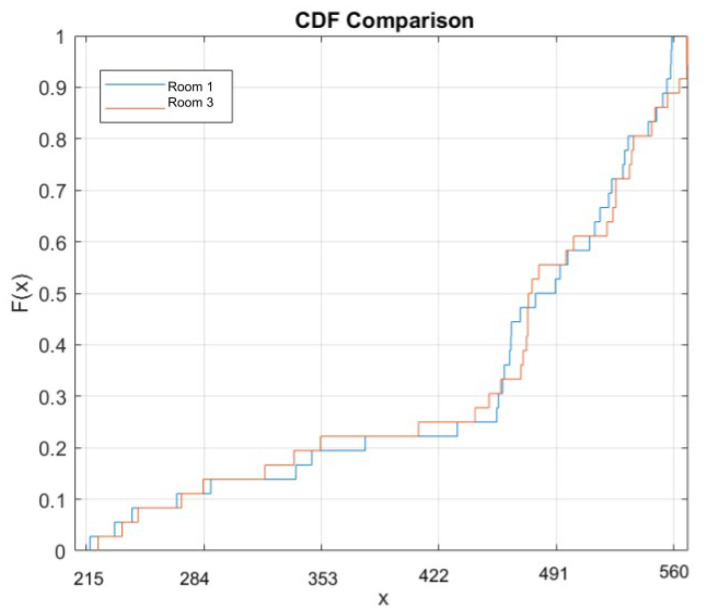
Cumulative density function (CDF) comparison between two rooms and normalized histograms (see Figure 11D,E, which illustrate the similarity in KS scores, which indicates their likelihood of coming from the same distribution).

**Figure 13 tomography-11-00025-f013:**
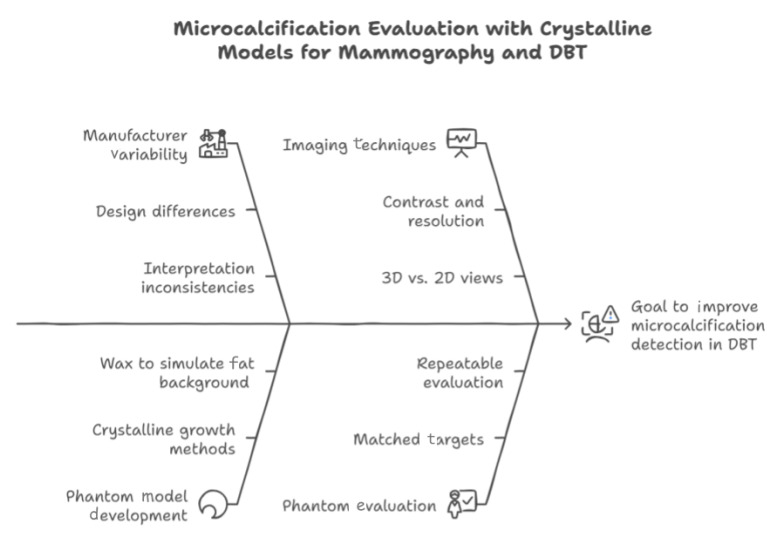
Visual overview of the core concepts of this study.

**Table 1 tomography-11-00025-t001:** Differences between vendors and various factors used in their mammography scanners.

Company Name	Fujifilm Medical Systems U.S.A. Inc.	GE Healthcare	Hologic	Siemens Healthineers
Scan angle	15 degrees	25-degree sweep angle for DBT	15 degrees	50 degrees
Matrix, pixels	24 × 30 cm: 4728 × 5928 pixels	2850 × 2394 pixels	3328 × 4096	2816 × 3584
Reconstruction style type (CCD, CsI, aSe)	aSe	CsI scintillator, single-piece construction	aSe	aSe
Reconstruction style	ISR (Iterative Super Resolution)	Iterative reconstruction algorithm called ASiR^DB^	FBP	FBP with iterative optimizations
Focal spot size, mm	0.3 mm	0.3 mm with a high-ratio grid (11:1)	0.3 mm	0.15/0.3 mm

Differences between imaging qualities amongst mammography vendors that arise from various factors, including technologically differing approaches to 3D digital breast tomosynthesis configurations, such as scan or sweep angles. These differences underscore the need for a standardized comparison target that includes advanced phantoms to ensure consistency in image quality.

## Data Availability

Available upon request.

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
