# Peer review of "A Novel Phantom for Standardized Microcalcification Detection Developed Using a Crystalline Growth System"

_tomography, 2025, doi:10.3390/tomography11030025_

Round 1

Reviewer 1 Report

Comments and Suggestions for Authors

Wu et al. present an interesting manuscript within the scope of this Journal. However, the manuscript has serious points, both in form and depth.

-The title should be revised. It does not correspond to what was presented in the manuscript.

-The introduction is too long, it seems more like an extensive review. It should be revised better and focus on the topic to justify the novelty of the study.

-The methodology is poorly described. The authors should make an effort to improve the reproducibility of the study.

-The results should be better described. The figures should be better placed and described in more detail.

-The discussion is simple. The authors should carry out a scientific discussion, with specialized references.

-A conclusions section should be included, with a clear translational component.

-A graphic summary is necessary.

Reviewer 2 Report

Comments and Suggestions for Authors

The manuscript presents a metodology for manufacturing a breast phantom including microcalcification clusters. Alhtough the topic is of interest, the manuscript presents some weakness that should be fixed before the pubblication. First, authors should focus and clarify the aim of the manuscript (producing a phantom?) and strees reasons related to the use of the proposed methodology compared to those in literature. The section realted to the restults is quite small, and mostly does not present quantitative ealutions. On the other hand, the introduction is too long and usually out of the scope of the paper. The English is sometimes poor and should be revised. on following specific comments:

Title: “Improving Digital Breast Tomosynthesis:” is not in line with the article aim. Hence, the article presents a phantom to be used in evaluations, not in improving the scanner. This part of the title must be removed.

Abstract – line 11: the focal spot size is fixed among all the manufacturers worldwide at 0.300 mm (0.100 in magnification mammography).

Abstract – line 12: “This study…” here there is a lack of the language. Please consider rephrase. It is not clear what is the aim.

Page 2 – line 74-79: I do not think that the aim of this work is to improve “methods of detection” as here kind of meant. I suggest the authors to clarify that the aim is to develop a breast phantom for image quality evaluations and comparison between scanners.

The introduction is too long and does not focus on the background and aim of the work. As example, sect. 1.1. seems out of the focus of the work, as the first part of 1.2. The rest should be rewritten to clarify what is the real aim of the work and what solves. Is it presenting a new breast phantom for evaluating the visibility of microcalcifications? In particular it should be highlighted what the introduced approach have more than literature. Authors can also take a look at recent literature reviews on the topic looking for something related (examples: DOI: 10.1016/j.ejmp.2020.11.025, DOI: 10.1016/j.radphyschem.2022.110715)

Page 10: it is not clear how authors mimicked different glandular fraction. Please clarify. Additionally, the benefits in using larger angular scan (or DBT in place of DM) is mostly related to the reduction of the anatomical noise due by the tissue superimposition. How authors modelled this anatomical noise?

Line 333 – It is not clear what are these models taken from BIRADS or VICTRE and of was then generate physical microcalcification on the basis of the images (fig. 8)

Caption of fig. 9 is too long. It should describe the figure. The rest goes into the text

Page 13 – line 400: Authors should explain how they compared the materials to the breast tissues? Just qualitatively?

Comments on the Quality of English Language

The language is too poor and need revision

Reviewer 3 Report

Comments and Suggestions for Authors

The article discusses a new form of manufacturing method for equating prostheses with ACR standard prostheses, and it has more realistic characteristics than standard prostheses.

Because the calcified area of this new type of prosthesis is indeed made of calcium carbonate, it may be more realistic than the current ACR clamp. However, the author needs to use a more scientific method to compare and describe the advantages of this new type of prosthesis compared with traditional ones. I only see a comparison in pictures, for instrument manufacturers, when they only use one type of prosthesis for evaluation, they cannot confirm whether your new prosthesis is more realistic.

Round 2

Reviewer 1 Report

Comments and Suggestions for Authors

 Accept in present form.

Author Response

We appreciate your comments. Thank you for your feedback.

Reviewer 2 Report

Comments and Suggestions for Authors

The manuscript remains qualitative. more quantitative evaluations are needed

Author Response

Please see attached comments
